# Synthesis and Biological Properties of Fluorescent Strigolactone Mimics Derived from 1,8-Naphthalimide

**DOI:** 10.3390/molecules29102283

**Published:** 2024-05-12

**Authors:** Ioana-Alexandra Bala, Alina Nicolescu, Florentina Georgescu, Florea Dumitrascu, Anton Airinei, Radu Tigoianu, Emilian Georgescu, Diana Constantinescu-Aruxandei, Florin Oancea, Calin Deleanu

**Affiliations:** 1Bioproducts Team, Bioresources Department, National Institute for Research & Development in Chemistry and Petrochemistry—ICECHIM, Splaiul Independenței Nr. 202, Sector 6, 060021 Bucharest, Romania; ioana.bala@icechim.ro; 2Faculty of Biotechnologies, University of Agronomic Sciences and Veterinary Medicine of Bucharest, Bd. Mărăști Nr. 59, Sector 1, 011464 Bucharest, Romania; 3“Petru Poni” Institute of Macromolecular Chemistry, Romanian Academy, Aleea Grigore Ghica Voda Nr. 41-A, 700487 Iaşi, Romania; alina@icmpp.ro (A.N.); airineia@icmpp.ro (A.A.); tigoianu.radu@icmpp.ro (R.T.);; 4“Costin D. Nenițescu” Institute of Organic and Supramolecular Chemistry, Romanian Academy, Splaiul Independentei Nr. 202B, Sector 6, 060023 Bucharest, Romania; fdumitra@yahoo.com; 5Enpro Soctech Com., Str. Elefterie Nr. 51, Sector 5, 050524 Bucharest, Romania; florentina.georgescu@enpro.ro; 6Research Center Oltchim, St. Uzinei 1, 240050 Ramnicu Valcea, Romania; g_emilian@yahoo.com

**Keywords:** strigolactone heterocycles, organic synthesis, fluorescence lifetimes, quantum yield, phytopathogen, fungal growth inhibition, hyphal branching

## Abstract

Strigolactones (SLs) have potential to be used in sustainable agriculture to mitigate various stresses that plants have to deal with. The natural SLs, as well as the synthetic analogs, are difficult to obtain in sufficient amounts for practical applications. At the same time, fluorescent SLs would be useful for the mechanistic understanding of their effects based on bio-imaging or spectroscopic techniques. In this study, new fluorescent SL mimics containing a substituted 1,8-naphthalimide ring system connected through an ether link to a bioactive furan-2-one moiety were prepared. The structural, spectroscopic, and biological activity of the new SL mimics on phytopathogens were investigated and compared with previously synthetized fluorescent SL mimics. The chemical group at the C-6 position of the naphthalimide ring influences the fluorescence parameters. All SL mimics showed effects similar to GR24 on phytopathogens, indicating their suitability for practical applications. The pattern of the biological activity depended on the fungal species, SL mimic and concentration, and hyphal order. This dependence is probably related to the specificity of each fungal receptor–SL mimic interaction, which will have to be analyzed in-depth. Based on the biological properties and spectroscopic particularities, one SL mimic could be a good candidate for microscopic and spectroscopic investigations.

## 1. Introduction

Strigolactones (SLs) are carotenoid derivatives with multiple functions in plants [1,2]. Their initial function in the terrestrial plants seems to be related to rhizosphere communication [3]. In *Physcomitrella patens*, moss SL acts a neighbor signal, similar to bacterial quorum sensing signals [4]. *P. patens* SL defective mutants fail in the detection of neighboring colonies [4]. Similarly, pea (*Pisum sativum*) mutant, defective in SL biosynthesis, fails to detect neighboring plants [5]. Strigolactones are involved in the recruitment of the arbuscular mycorrhizal (AM) fungi [6,7,8] and symbiotic nitrogen-fixing rhizobia [9,10,11]. The difference in SL exudation between wild plants and SL-defective plant mutants shape differently the rhizosphere microbiome in *Arabidopsis thaliana* [12], soybean (*Glycine max*) [13], and rice (*Oryza sativa*) [14]. The rhizosphere SLs are also used as a cue for plant root presence by the parasitic plant seeds [15,16,17,18] and plant pathogenic fungi [19].

SLs have been recognized as a class of phytohormones involved in plant development [20,21,22,23,24,25,26,27,28,29,30] and in the orchestration of the plant responses to abiotic stresses [31,32,33]. The initial described natural SLs have complex structures containing a tricyclic lactone system (A, B, and C rings) connected through an enol-ether bond to a furan-2-one ring (D)—canonical SLs [34] (Figure 1).

In the last decade, natural SLs that do not have the specific tricyclic lactone system (A, B, and C rings) were discovered—the non-canonical SLs, with the bioactiphore and the butenolide D-ring but without the A-, B-, or a closed C-ring [35]. The synthetic SLs further expand the structural varieties, SL analogs, and SL mimics demonstrating specific SL activities.

The SL multifaceted roles, as endo- and exo-signals, are related to the presence of C (open or closed) and D rings, connected via an enol ether bridge. This part of the SLs (canonical or non-canonical) bind to similar receptors from the (super)family of α/β hydrolase fold proteins and trigger the cascade reactions that determine the specific response to SLs in different organisms [36].

Naturally occurring SLs are released from plants in tiny amounts. Complex structures and laborious synthetic procedures for the multi-gram amounts of natural SLs have raised interest in synthesizing simplified molecules that display various SL bioactivities. Therefore, synthetic SL analogs that maintain the bioactiphore CD part of natural SLs [37,38,39,40,41,42,43,44,45] were synthetized. These SLs analogs are structurally more related to canonical SLs. Simpler SL mimics, more structurally related to non-canonical SLs, were proposed. In these SL mimics where the methyl-substituted D-ring is directly linked to an arylthiol, aryloxy or aroyloxy moieties have been synthesized [43,46,47,48].

The list of synthetic SL analogs includes GR24, (3E,3aR,8bS)-rel-3-[[[(2R)-2,5-dihydro-4-methyl-5-oxo-2-furanyl]oxy]methylene]-3,3a,4,8b-tetrahydro-2*H*-indeno[1,2-*b*]furan-2-one, named after its inventor Gerald Rosebery, which was first synthesized in 1981 [38] and largely used as a strigolactone (strigol type) synthetic replacer in laboratory studies; GR7, with the A-ring removed from GR24 [38]; Nijmegen-1, tested in field conditions as a suicidal germination stimulant for *Orobanche ramosa* [37]; CISA-1, a fluorescent alternative to GR24 [49]; and PLN65, PLO65, and PLS6—conjugated fluorescent strigolactone analogs [50]. Recently published were analogs of non-canonical SLs, cannalactone analogs named (±)-SdL19 and (±)-SdL118, that stimulate the germination of *Phelipanche ramose* and exhibit other SL activities, e.g., the inhibition of plant shoot branching and stimulation of hyphal branching in AM fungus *Rhizophagus irregularis* [51]. Strigolactone mimics include 4BD, 4-bromodebranone [52]; PL02, *N*-(4-methyl-5-oxo-2,5-dihydrofuran-2-yl)-3-nitrophthalimide, and other similar phthalimide-derived compounds [53]; SL-6, 2-(4-methyl-5-oxo-2,5-dihydro-furan-2-yloxy)-benzo[*de*]isoquinoline-1,3-dione [54]; contalactone, i.e., contaminant lactone of GR24 [55]; and carlactonic acid, rac-/(R)-/(S)-(E,2E)-2-[(4-Methyl-5-oxo-2H-furan-2-yl)oxymethylene]-4-(2,6,6-trimethylcyclohexen-1-yl)but-3-enoic acid, and its methyl derivative [56]. A detailed list of SL analogs and mimics is presented in several reviews [28,57,58].

Structure–activity relationship data on SLs suggest the existence of different perception systems for their various biological activities. In plants, DWARF14 (D14), an α/β hydrolase, is the strigolactone main receptor [59], which evolved by duplication and differentiation from the karrikin (KAR) receptor KAI2 [3]. In parasitic plants, the SL perception as a cue is performed by a paralog of KAI2, α/β hydrolase ShHTLs/ShKAI2s [60,61]. In fungal plant pathogen *Cryphonectria parasitica*, CpD14, a structural homolog of plant receptor D14, is involved in the SL response. Many fluorescent SL analogs or mimics, most of which bear known fluorophore molecules, have already been prepared to reveal all protein receptors and the distribution of SLs in plants, and to elucidate their mode of action [49,50,62,63,64,65,66].

Continuing our research on bioactive compounds [54,67,68,69], we have been interested in developing new bioactive and fluorescent SL mimics suitable for bio-imaging studies in fungi and/or plant cells. Considering the potential exploitation of SL mimics as active ingredients of the plant biostimulants, due to their direct effect on plant tolerance to abiotic stress, and especially to drought [31,70,71] and/or mediators of plant interaction with beneficial and/or detrimental rhizosphere organisms for plants, as a component of next-generation plant biostimulants [72], we developed new fluorescent SL mimics derived from the 1,8-naphthalimide fluorophore molecule. The 1,8-Naphthalinimide ring system is the central component of different fluorescent compounds with potential applications in various fields [73,74,75,76,77,78,79]. We already reported an SL mimic bearing the 1,8-naphthalimide system, namely 2-(4-methyl-5-oxo-2,5-dihydro-furan-2-yloxy)-benzo[*de*]isoquinoline-1,3-dione **1** (Figure 2), and its spectroscopic and biological characteristics [54,80].

We reported other, new fluorescent SL-mimics, with the amino-substituted 1,8-naphthalimide ring system, namely 2-(4-methyl-5-oxo-2,5-dihydrofuran-2-yloxy)-6-(4-methylpiperidin-1-yl)-benzo[*de*]isoquinoline-1,3-dione **2** and 2-(4-methyl-5-oxo-2,5-dihydro-furan-2-yloxy)-6-(4-benzylpiperidin-1-yl)-benzo[*de*]isoquinoline-1,3-dione **3** (Figure 1) [81]. They have been prepared via successive reactions of 4-chloro-1,8-naphthalic anhydride **4** with 4-methylpiperidine and 4-benzylpiperidine, respectively; reactions of the resulting 4-amino-1,8-naphthalic anhydrides with hydroxylamine hydrochloride to give the fluorescent 2-hydroxy-6-amino-1,8-naphthalimide intermediates [79]; and condensations of 2-hydroxy-6-amino-1,8-naphthalimide intermediates with 5-bromo-3-methyl-5*H*-furan-2-one 6 to access fluorescent SL mimics **2** and **3**, respectively [81].

Interesting biological and fluorescent properties of the previously reported SL mimics bearing a 1,8-naphthalimide system [54,80,81] prompted us to prepare new, potentially bioactive and fluorescent SL mimics aiming to develop SL-mimics-based agricultural inputs that modulate plant interactions with fungal pathogens. Thus, two other new SL mimics containing halo-substituted 1,8-naphthalimide molecules connected through an ether link to a bioactive furan-2-one unit have been prepared and fully characterized. They have been used in an alternative synthetic procedure towards the previously reported SL mimics **2** and **3**, respectively. The biologic properties of both the growth inhibition of phytopathogenic fungi and the effects on the hyphal branches have been investigated for all these SL mimics.

## 2. Results and Discussion

### 2.1. Synthesis of 1,8-Naphthalimide-Derived Strigolactone Mimics

The synthesis of the potentially bioactive and fluorescent SL mimics started from commercially available 4-chloro-1,8-naphthalic anhydride **4**. The reactions of **4** with hydroxylamine hydrochloride in dioxane led to 2-hydroxy-6-chloro-1,8-naphthalimide intermediate **5** [79]. The condensation reaction of the resulting 2-hydroxy-6-chloro-1,8-naphthalimide intermediate **5** with common intermediate 5-bromo-3-methyl-5*H*-furan-2-one **6** carried out in DMF, using anhydrous potassium carbonate as the basic catalyst, gave a new SL mimic, namely 6-chloro-2-(4-methyl-5-oxo-2,5-dihydrofuran-2-yloxy)-benzo[*de*] isoquinoline-1,3-dione **7** (Figure 1). The common intermediate in the SL mimic syntheses, i.e., 5-bromo-3-methylfuran-2-one **6**, was prepared by brominating the commercially technical product 3-methyl-5*H*-furan-2-one with *N*-bromosuccinimide (NBS), in the presence of a radical initiator [82]. Based on the new SL mimic **7**, we developed a new synthetic route towards previously reported putative fluorescent SL mimics **2** and **3** [81]. The two putative fluorescent SL mimics **2** and **3** were easily prepared by the coupling reactions of SL mimic **7** with cyclic secondary amines 4-methylpiperidine and 4-benzylpiperidine, respectively. These reactions took place in *N*-methylpyrrolidone (NMP), in the presence of triethylamine, at the reflux temperature of the reaction mixture. In this way, we developed a new synthetic protocol which starts from commercially available 4-chloro-1,8-naphthalic anhydride **4** and gave easier access to 1,8-naphthalimide-based SL mimics **7**, **2**, and **3** (Figure 1).

A new potentially bioactive SL mimic, namely 6-bromo-2-(4-methyl-5-oxo-2,5-dihydrofuran-2-yloxy)-benzo[*de*] isoquinoline-1,3-dione **10** (**SL-27**), was also easily prepared via the reaction of commercially available 4-bromo-1,8-naphthalic anhydride **8** with hydroxylamine hydrochloride in pyridine [83] to obtain 2-hydroxy-6-bromo-1,8-naphthalimide intermediate **9**, followed by the coupling reaction of intermediate **9** with 5-bromo-3-methylfuran-2-one intermediate **6** (Figure 2). The yields in 2-hydroxy-6-halo-1,8-naphthalimide intermediate **8** and **9**, respectively, were similar (about 90%). The two fluorescent SL mimics **2** and **3** were again easily prepared by the coupling reactions of SL mimic **10** with 4-methylpiperidine and 4-benzylpiperidine, respectively, in N-methylpyrrolidone at reflux temperature, in the presence of triethylamine. In this way, starting from commercially 4-bromo-1,8-naphthalic anhydrides **8**, we gained access to 1,8-naphthalimide-based SL mimics **10**, **2**, and **3**. The final yields in SL mimics were similar irrespective of whether the synthetic procedure started from 4-chloro- or 4-bromo-1,8-naphthalic anhydride.

The structures of all synthesized compounds were proven by ^1^H, ^13^C, and ^15^N NMR spectroscopy. The 1,8-naphthalimide and 3-methylfuran-2-one fragments have individual resonance signals, readily assigned based on the bond correlations from bidimensional homo- and heteronuclear experiments. For example, in the ^1^H NMR spectrum corresponding to SL mimic **7**, presented in Figure 3, there are five signals in the low-field region (above 8.00 ppm) assigned to the naphthalimide fragment as follows: 8.03 ppm doublet of doublets H-8, 8.05 ppm doublet H-5, 8.45 ppm doublet H-4, 8.61 ppm doublet of doublets H-9, and 8.63 ppm doublet of doublets H-7. The other fragment, 3-methylfuran-2-one, has three characteristic signals at 1.91 (triplet, CH_3_), 6.60 (quintet, H-2′), and 7.45 ppm (quintet, H-3′).

Long-range proton–proton couplings, over four and five bonds, were observed in H,H-COSY (Figure 4a) between methyl protons and the two furan cycle protons. These couplings, facilitated by the double bond, are responsible for the unexpected signal shapes: triplet for CH_3_ and quintet for the two CH groups.

The successful formation of the naphthalimide intermediate and its consequent condensation with 5-bromo-3-methyl-5H-furan-2-one were demonstrated from long-range proton–nitrogen interactions—Figure 4. In the H,N-HMBC spectrum corresponding to SL mimic **7** (Figure 4b), a correlation signal was observed between the nitrogen atom from 224 ppm and the furan cycle proton from 6.60 ppm.

The exact molecular weights were confirmed by high-resolution mass spectrometry and the consistency of the molecular formulas was assessed by comparison of the experimental and simulated isotopic patterns. The proton and carbon NMR assignments and coupling constants are presented in the experimental section. All NMR spectra together with the experimental and simulated MS isotopic patterns are presented as Appendix A).

The electronic absorption spectra of the 1,8-naphthalimide-based strigolactone mimics **7** and **10** in solvents with different polarities such as toluene (PhMe, ε = 2.38), dichloromethane (DCM, ε = 9.08), and *N*,*N*-dimethylformamide (DMF, ε = 38.25) were investigated. The spectral parameters for these compounds, namely absorption maxima (λ_max_), emission maxima (λ_em_), and Stokes shifts (Δν), are listed in Table 1. In Appendix A, the electronic absorption and, respectively, emission spectra in dichloromethane are presented for SL mimics **7** and **10** (see Appendix A).

SL mimics **1**, **7**, and **10** present absorption bands in the range of 300–370 nm in DCM (Table 1) with maxima centered around 335 nm (**1**), 343.5 nm (**7**), and 345 nm (**10**), respectively, arising from a π-π* electron transition of the 1,8-naphthalimide ring [75,76,77,78,79,80,81]. Expectably, the introduction of halogen groups in the C-6 position of the naphthalimide ring (SL mimics **7** and **10**) determines a red shift in the absorption maximum as compared to the unsubstituted SL mimic **1**. The solvent polarity has a small effect on the absorption maxima of SL mimics **7** and **10** (Table 1). The SL mimics containing the piperidine substituent at the 1,8-naphthalimide unit in the C-6 position (SL mimics **2** and **3**) display the longest-wavelength absorption in the visible range without a clear vibrational structure [81]. Changing the solvent polarity from nonpolar toluene to polar DMF, a red shift in the position of the absorption band of 11 nm was found and this red shift was 15 nm in DCM for **2**.

The fluorescence characteristics of these SL mimics are displayed in Table 1. In DMF solutions, SL mimics **7** and **10** show an intense violet emission at about 400 nm. From Table 1, it is noticed that the influence of the solvent polarity on the fluorescence maxima is more pronounced than that on the electronic absorption spectra. The Stokes shift (Δ*ν*) reveals the difference in the structure and properties of fluorophores between the ground and the first excited state, estimated according to Equation (1):(1)$$∆\nu =\nu_{abs}-\nu_{em}=\left(\frac{1}{\lambda_{abs}}-\frac{1}{\lambda_{em}}\right)\times 10^{7}$$

The values Δ*ν* for SL mimics under study are in the 3200–9500 cm^−1^ range in the three solvents, which are in agreement with the values reported for other naphthalimide derivatives [75,78,81]. For SL mimics **7** and **10**, a hypsochromic shift in emission maxima was induced when the solvent was changed from toluene to DMF. This blue shift can indicate that a decrease in the electron density can occur in the excited state relating to the ground state.

The photophysical parameters of these derivatives such as quantum yield and fluorescence lifetimes are listed in Table 1 and Table 2. It is observed that the values of the fluorescence quantum yield depend significantly on the solvent nature.

SL mimic **7** presents moderately high values of quantum yield in toluene (5.49%) and DCM (6.82%), whereas in DMF, a very small value was obtained (0.01%). Replacing chlorine in C-6 of the 1,8-naphthalimide ring with bromine determined an evident decrease in the quantum yield value for SL mimic **10** in toluene (0.65%) and DCM (1.57%), but a slight increase in DMF (0.89%).

The emission profiles were fitted to one, two, or three exponential curves, depending on the derivative structure and solvent. The fluorescence decay traces of naphthalimide derivatives **1**, **7**, and **10** are displayed in Appendix A. From the fitted decay parameters, the decay profile of **7** exhibited two lifetime components in toluene with a predominance of the short lifetime around 0.94 ns and an amplitude of 82.14%, whereas in DMF, a monoexponential decay was noticed with a very short lifetime. The bromine derivative **10** presented a biexponential decay in DMF, leading to emission lifetimes of 0.33 ns (53.86%) and 4.99 ns (46.14%), respectively, whereas in DCM, a triexponential decay was observed (Table 2). The increase in solvent polarity leads to a decrease in the fluorescence lifetime in DMF, which was correlated with the changes in the emission quantum yield. It must be mentioned that the highest values of the fluorescence quantum yield were obtained for 1,8-naphthalimide derivatives with a piperidine substituent at the C-6 position of the naphthalimide ring (SL mimics **2** and **3**) in nonpolar solvents (see Appendix A) [81]. Consequently, it would be suitable to investigate their temporal distribution *in planta* or in fungal organisms or to study the activity of SL receptors in vitro [84,85,86,87] after the hydrolysis of the bioactive 3-methyl-5*H*-furan-2-one unit.

### 2.2. Biological Activity of Synthesized Compounds

The influence of the 6-substituted 1,8-naphthalimide-derived SL mimics **2**, **3**, **7**, and **10** (Figure 3) on radial fungal growth and hyphal branching was investigated.

For presenting the biological activity, we shall continue to use the final given names for the synthesized SL mimics, i.e., SL20 for mimic **2**, SL21 for mimic **3**, SL26 for mimic **7**, and SL27 for mimic **10**. These given names are essential for reporting further tests with these molecules. We monitored the SL dose dependence by testing three SL concentrations: 5 × 10^−6^ M (C1), 10^−5^ M (C2), and 5 × 10^−5^ M (C3). The reference compound was GR24, an SL analog with a proven effect on the branching of plant pathogenic fungi [19,88]. This SL analog was developed from strigol as a basic blueprint and includes a tricyclic lactone system (A, B, and C rings) connected through an enol-ether bond to a furan-2-one ring (D) [38]. The effects were tested on the following fungal strains: *Fusarium graminearum* DSM 4527, *Rhizoctonia solani* DSM 22842, *Sclerotinia sclerotium* DSM 1946, and *Colletotrichum acutatum* CBS 113008.

Based on the experimental results, it was observed that both GR24 and the new SL mimics had a statistically significant inhibitory effect on the growth of all four phytopathogenic fungi that were tested (Figure 5).

As can be observed in Figure 5, even at the lowest concentration tested of 5 × 10^−6^ M (C1), the inhibitory effect was noticeable. The inhibitory effects on fungal colonies were dependent on both the fungal strain and the SL mimic tested. The compounds showed a dose-dependent suppression of the fungal strains. All SL mimics inhibited *F. graminearum* compared with the acetone control. Except SL21, all the other SL mimics had a stronger inhibitory effect than GR24, with SL27 being the strongest inhibitor. In the case of *R. solani*, all SL mimics inhibited the fungal growth except SL21 at C1. The inhibition by the SL mimics was comparable to GR24, except SL21, which had a lower effect. The effects of SL20 and SL21 on *S. sclerotium* were similar to those of GR24, but SL26 and SL27 had slightly lower effects than GR24. In the case of *C. acutatum*, all SL mimics had a similar or even higher inhibitory effect than GR24 on fungal growth.

Taken together, all the newly synthesized SL mimics had a similar or even higher effect on the phytopathogens tested, especially at the lowest compound concentration applied. The effects depend on both the strain and the SL mimic. The SL mimics differ by the chemical group in the C-6 position of the naphthalimide ring, and it is intriguing that this can influence the effects observed, in a strain-dependent manner. Future work should aim to bring light to the mechanistic aspects.

We next investigated the effects of the SL mimics on the hyphal branching of phytopathogens, in comparison with GR24. Appendix A show the typical hyphae development of *F. graminearum*, *R. solani*, *S. sclerotium*, and *C. acutatum*, respectively, from our experimental treatments. In all cases, the number of branched hyphae was recorded along the length of 2000 µm of the primary hyphae, starting from the end of the youngest hyphal edge. Since many primary hyphae were not straight, in these cases, the total distance consisted of several shorter segments along the length of the linear hyphae.

In Figure 6, the pattern of hyphal branching and analysis for each of the four fungi show hyphal branching up to the 5th order, but most of the variants displayed maximum 4th-order branching.

The quantification of each type of branch for all four phytopathogens and variants is shown in Figure 7. For *F. graminearum,* the vast majority of the treatments with SL mimics were significantly different from the control in the case of the second-order branches except SL21 at the lowest concentration C1. Most treatments induced a higher number of second-order branches, except SL21 C2 and C3, which slightly inhibited the number of branches compared with the control. GR24, SL20, and SL27 stimulated the third-order branches, whereas SL21 inhibited and SL26 slightly stimulated them only at the highest concentration C3. In the case of the fourth-order branches, only SL20 and SL27 induced a statistically significant appearance compared with the control (Figure 7a).

It is worth mentioning that the dependence on the SL dose was specific for each SL mimic: whereas the GR24 and SL27 stimulatory and SL21 inhibitory effects increased with the concentration (SL27 showed signs of saturation at C2 and C3), SL26 did not show a concentration dependence. The most intriguing dependence was that of SL20, which had a heterogeneous behavior: the 2nd-order branching increased with the concentration, the 3rd-order branching had a maximum at C2, and the 4th-order branching decreased with the concentration (Figure 7a). This suggests that, in the case of SL20, the higher the branching order, the lower the optimum SL mimic concentration for maximum branching that is needed. This aspect, as well as the inhibitory effect observed for SL21, deserves further investigation.

For *R. solani*, the second-order branching was stimulated only by GR24 in a dose-dependent manner and by SL20 at the lowest concentration C1, after which the C2 and C3 concentrations had control-like effects. SL21 did not have a significant effect, whereas SL26 and SL27 slightly inhibited the second-order branching. The 3rd- and 4th-order branching was stimulated only by GR24. SL20 at C1 had no effect and all the other variants inhibited the third-order branching. All SL mimics inhibited the fourth-order branching. We underline the observation that *R. solani* had a significant number of 3rd- and 4th-order branching in the control, much higher than the other strains (Figure 6b). It is not clear whether the inhibitory effects of the SL mimics are related to this aspect, and this needs more in-depth studies.

Almost all SL variants stimulated the 2nd- and 3rd-order branching of *S. sclerotium*. SL26 C3 and SL27 C2 and C3 did not have a stimulatory effect on the third-order branching. In general, it was observed that lower SL concentrations had a slightly higher branching effect, with the SL20 effect being higher than the GR24 effect. The fourth-order branching was absent in the control with acetone and appeared when GR24 or SL mimics were applied, but the difference was not statistically significant (Figure 6c).

For *C. acutatum*, all treatments, except SL26 C3, induced significantly higher second-order branching than the control. In the case of the third-order branches, only the treatments with GR24, SL20 C1, C2, and SL27 C3 had a stimulatory effect. The other treatments inhibited or, in a few cases, did not have any effect. The fourth-order branches were slightly stimulated only by GR24 (Figure 6d).

As in the case of fungal growth inhibition, the effects on the hyphal branches is strain- and SL-mimic-dependent. Moreover, whereas the concentration dependence of fungal growth inhibition is homogeneous, the number of hyphal branches varies in a hormetic manner and the effects depend on the hyphal order as well. The observation that the chemical group in the C-6 position of the naphthalimide ring influences this pattern is in accordance with the relatively recent paradigm shift by which it was proposed that the signal transduction induced by the SL receptor is triggered by the conformational changes in the complex with the intact SL, and not by the butenolide molecule released upon hydrolysis [89]. This variation in structure could induce a variation in the receptor–SL mimic affinity as well as variations in the conformational changes, which could explain the heterogeneity in the biological outcome observed. Recently, a fungal homolog from *Cryphonectria parasitica*, of the plant DWARF14 (D14) SL receptor, was reported and characterized [90]. This receptor is able to hydrolyze GR24 and this enzymatic activity requires the known catalytic triad Ser/His/Asp. The knockout of this gene was reported to reduce the fungus response to SLs. We believe that the specificity of the specie–SL mimic pair is probably based on the differences in the binding site of the fungal receptor among species. Each SL will have to be taken separately and investigated in-depth in order to understand the differences observed and optimize the desired response, which we plan to do in the future. From the analysis and comparison of all data, the most promising candidate for future fluorescence studies seems to be SL20: it behaves similarly to GR24 for all the strains tested and it has the highest fluorescence quantum yield and a reasonable lifetime even in the more polar solvent, DMF, important for biological imaging applications where a polar environment is involved.

## 3. Materials and Methods

### 3.1. Materials

The starting materials 4-chloro-1,8-naphthalic anhydride **4** and 4-bromo-1,8-naphthalic anhydride **8** were commercial products (Sigma-Aldrich, St. Louis, MO, USA). The intermediate compounds, 6-chloro-2-hydroxy-benzo[*de*]isoquinoline-1,3-dione **5** [65] and 6-bromo-2-hydroxy-benzo[*de*]isoquinoline-1,3-dione **9**, respectively [69], were prepared according to the reported procedures. The common intermediate, 5-bromo-3-methyl-5*H*-furan-2-one **6**, was obtained in good yield by the bromination of 3-methyl-5*H*-furan-2-one with *N*-bromosuccinimide in CCl_4_ in the presence of small amounts of benzoyl peroxide or azobisisobutyronitrile [69]. All reagents and solvents, including the solvents of spectrophotometric degree utilized in this work, were commercial products (Sigma-Aldrich, St. Louis, MO, USA) and were used without purification. The fungal strains tested were represented by *C. acutatum* CBS 113008 (Culture collection of fungi and yeasts, Westerdijk Fungal Biodiversity Institute, Utrecht, The Netherland), *F. graminearum* DSM 4527, *R. solani* DSM 22842, and *S. sclerotium* DSM 1946 (DSMZ, Braunschweig, Germany). For SL mimic solubilization and strain culture, the following commercial reagents were used: GR24 racemic (Strigolab, Turin, Italy), acetone (Chimreactiv, Bucharest, Romania), potato dextrose agar, and agar (Scharlau, Barcelona, Spain).

### 3.2. Structural and Physical–Chemical Characterization of the Synthesized Compounds

A Boetus apparatus was used to determine the melting points of the synthetic strigolactone mimics. These melting points are uncorrected. For recording the FTIR spectra in KBr pellets, the Nicolet Impact 410 spectrometer (Thermo Scientific, Waltham, MA, USA) was used. We used a Bruker Avance Neo spectrometer for the NMR analyses (Bruker Biospin, Ettlingen, Germany), operating at 600.1, 150.9, and 60.8 MHz for ^1^H, ^13^C, and ^15^N, respectively. The 1D and 2D spectra were recorded on a 5 mm multinuclear inverse detection z-gradient probe. Chemical shifts are reported in δ units (ppm) and reference the residual solvent signals (^1^H at 2.51 ppm and ^13^C at 39.4 ppm). The ^15^N chemical shifts reference liquid ammonia (0.0 ppm) using nitromethane (380.2 ppm) as the external standard. The 2D NMR homo- and heteronuclear correlations were used to make the unambiguous 1D NMR signal assignments. H,H COSY, H,C HSQC, and H,C HMBC experiments were recorded using standard pulse sequences in the version with z-gradients, as delivered by Bruker with TopSpin 4.0.8 spectrometer control and processing software. The ^15^N chemical shifts were obtained as projections from the 2D indirectly detected H,N HMBC spectra, employing a standard pulse sequence in the version with z-gradients as delivered by Bruker (TopSpin 4.0.8).

High-resolution MS spectra recorded on a Bruker Maxis II QTOF spectrometer (Bruker Daltonics, Bremen, Germany), with electrospray ionization (ESI) in the positive mode, were used to obtain the exact molecular weights and isotopic patterns.

Electronic absorption spectra were performed with a Specord 210PLUS UV-Vis spectrometer (Analytik Jena, Jena, Germany) in 10 mm path length cuvettes.

The fluorescence spectra were recorded on a Perkin Elmer LS55 luminescence spectrometer (Perkin Elmer, Inc., Waltham, MA, USA) using quartz cells of 10 mm optical path length. Time-correlated single-photon-counting measurements were performed using an FLS980 fluorospectrometer (Edinburgh Instruments, Livingston, UK), with 10 mm quartz cuvettes. A nanosecond diode laser centered at 375 nm was used as the excitation source. The fluorescence quantum yield (Φ) was determined in dilute solutions (A < 0.1) at the excitation wavelength corresponding to the maximum of the absorption band. Fluorescence decay measurements were carried out at the emission maxima. The decay data were fitted to the multiexponential model (1) to obtain the fluorescence lifetimes. In Equation (2), *I*(*t*) denotes the emission intensity at a time *t*, and *a_i_* and *τ_i_* represent the pre-exponential factor and the decay time of component *i*, respectively. The decay times and pre-exponential factors were estimated by minimizing the reduced chi-square χ^2^ value and by the plots of the weighted residuals versus channel number. All curve fittings possess χ^2^ values below 1.1 and a symmetrical distribution of the residuals.
(2)$$I(t)=\sum_{i}a_{i}\exp\left(-\frac{t}{\tau_{i}}\right)$$

### 3.3. Synthesis of 6-Chloro-2-(4-methyl-5-oxo-2,5-dihydro-furan-2-yloxy)-benzo[de]isoquinoline-1,3-dione (***7***) ***SL-26*** or 6-Bromo-2-(4-methyl-5-oxo-2,5-dihydro-furan-2-yloxy)-benzo[de]isoquinoline-1,3-dione (***10***) ***SL-27***

An amount of 15 mL of DMF K_2_CO_3_ (2.76 g, 20 mmol) was added to a solution of 6-chloro-2-hydroxy-benzo[*de*]isoquinoline-1,3-dione **5** (2.475 g, 10 mmol) or 6-bromo-2-hydroxy-benzo[*de*]isoquinoline-1,3-dione **9** (2.921, 10 mmol), followed by the slow addition of crude 5-bromo-3-methyl-5*H*-furan-2-one **6** (2.21 g, 12.5 mmol) under stirring at room temperature. The reaction mixture was stirred at room temperature for 24 h. Next, the reaction mixture was poured into 100 mL of water and extracted with 3 × 100 mL of CHCl_3_; the combined extracts were washed with an equal volume of water and dried on anhydrous Na_2_SO_4_. The solvent was partly removed under reduced pressure and the solid formed was filtered and recrystallized.

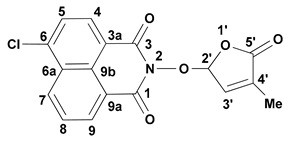

*6-Chloro-2-(4-methyl-5-oxo-2,5-dihydro-furan-2-yloxy)-benzo[de]isoquinoline-1,3-dione (***7***)* **SL-26**. Light beige crystals, m.p. 215–217 °C (CHCl_3_/MeOH). Yield 67% (2.3 g). Anal. Calcd. for C_17_H_10_ClNO_5_ (343.73): C, 59.40; H, 2.93; N, 4.07%. Found: C, 58.57; H, 3.02; N, 3.95%. FTIR (KBr, ν_max_): 2922, 1777, 1727, 1684, 1589, 1367, 1234, 1192, 1090, 1029 cm^−1^. ^1^H NMR (600 MHz, DMSO-d_6_), δ (ppm): 1.93 (t, *J* = 1.4 Hz, 3H, CH_3_), 6.62 (quintet, *J* = 1.4 Hz, 1H, H-2′), 7.45 (quintet, *J* = 1.4 Hz, 1H, H-3′), 8.05 (dd, *J* = 8.3, 7.4 Hz, 1H, H-8), 8.08 (d, *J* = 7.9 Hz, 1H, H-5), 8.45 (d, *J* = 7.9 Hz, 1H, H-4), 8.63 (dd, *J* = 7.3, 1.0 Hz, 1H, H-9), 8.66 (dd, *J* = 8.4, 1.0 Hz, 1H, H-7). ^13^C NMR (150.9 MHz, DMSO-d_6_), δ (ppm): 10.2 (CH_3_), 104.0 (CH-2′), 121.8 (C-3a), 123.1 (C-9a), 127.9 (CH-5 and C-9b), 128.6 (C-6a), 128.8 (CH-8), 130.7 (CH-7), 131.2 (CH-4), 132.0 (CH-9), 134.3 (C-4′), 138.1 (C-6), 141.7 (CH-3′), 159.7 (CO-3), 159.9 (CO-1), 170.9 (CO-5′). ^15^N NMR (60.8 MHz, DMSO-d_6_), δ (ppm): 223.9 (N-2). HRMS-ESI (*m*/*z*): [M + Na]^+^ for C_17_H_10_ClNNaO_5_, calcd. 366.0140, found 366.0150.

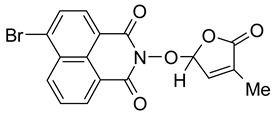

*6-Bromo-2-(4-methyl-5-oxo-2,5-dihydro-furan-2-yloxy)-benzo[de]isoquinoline-1,3-dione (***10***).* **SL-27**. Light beige crystals, m.p. 238–241 °C (MeNO_2_). Yield 65% (2.5 g). Anal. Calcd. for C_17_H_10_BrNO_5_ (388.18): C, 52.60; H, 2.60; N, 3.61%. Found: C, 51.89; H, 2.70; N, 3.51%. ATR-FTIR (solid, ν_max_): 1775, 1721, 1679, 1579, 1359, 1336, 1229, 1094, 1028 cm^−1^. ^1^H NMR (600 MHz, DMSO-d_6_), δ (ppm): 1.93 (bs, 3H, CH_3_), 6.62 (bs, 1H, H-2′), 7.48 (bs, 1H, H-3′), 8.03 (t, *J* = 7.8 Hz, 1H, H-8), 8.26 (d, *J* = 7.8 Hz, 1H, H-5), 8.37 (d, *J* = 7.8 Hz, 1H, H-4), 8.59 (d, *J* = 8.5, 1H, H-9), 8.62 (d, *J* = 7.2 Hz, 1H, H-7). ^13^C NMR (150.9 MHz, DMSO-d_6_), δ (ppm): 10.2 (CH_3_), 104.1 (CH-2′), 122.4 (C-3a), 123.1 (C-9a), 127.8 (C-9b), 129.0 (CH-8), 129.9 (C-6), 130.0 (C-6a), 131.4 (CH-4), 131.5 (CH-5), 132.0 (CH-7), 133.3 (CH-9), 134.3 (C-4′), 141.8 (CH-3′), 159.9 (CO-3), 160.0 (CO-1), 170.9 (CO-5′). ^15^N NMR (60.8 MHz, DMSO-d_6_), δ (ppm): 223.0 (N-2). HRMS-ESI (*m*/*z*): [M + Na]^+^ for C_17_H_10_BrNO_5_, calcd. 409.9640, found 409.9655.

### 3.4. New Synthetic Procedure for SL Mimics ***2*** and ***3***

An amount of 30 mL of *N*-methylpyrrolidone, 4-methylpiperidine, and 4-benzylpyperidine (12 mmol), respectively, or 2.1 mL (15 mmol) of triethylamine were added to a suspension of SL mimic **7** (2.06 g, 6 mmol) or SL mimic **10** (2.33 g, 6 mmol), and the reaction mixture was heated at reflux temperature for 24 h. The reaction mixture was poured in 100 mL of water and extracted with 3 × 150 mL of CHCl_3_, and the combined extracts were washed with an equal volume of water and dried on anhydrous Na_2_SO_4_. The solvent was distilled off, and the solid was filtered and crystallized to obtain SL mimics **2** and **3**, respectively.

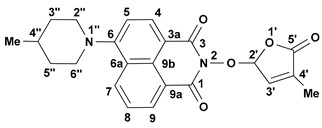

*2-(4-Methyl-5-oxo-2,5-dihydrofuran-2-yloxy)-6-(4-methylpiperidin-1-yl)-benzo[de]isoquinoline-1,3-dione (***2***):* **SL-20.** Orange crystals, m.p. 198–200 °C (CHCl_3_). Yield: 60% (1.45 g). Anal. Calcd. for C_23_H_22_N_2_O_5_ (406.44): C, 67.97; H, 5.46; N, 6.89%. Found: C, 68.10; H, 5.54; N, 5.93%. FTIR (KBr, ν_max_): 3432, 2916, 1784, 1715, 1676, 1584, 1457, 1363, 1229, 1193, 1088 cm^−1^. ^1^H NMR (600 MHz, DMSO-d_6_), δ (ppm): 1.04 (d, *J* = 6.4 Hz, 3H, CH_3_-4″), 1.52 (dd, *J* = 22.6, 11.2 Hz, 1H, CH_2_-3″A), 1.63–1.65 (m, 1H, CH-4″), 1.82 (d, *J* = 11.9 Hz, 1H, CH_2_-3″B), 1.92 (bs, 3H, CH_3_-4′), 2.93 (t, *J* = 11.9 Hz, 1H, CH_2_-2″A), 3.56 (d, *J* = 11.9 Hz, 1H, CH_2_-2″B), 6.60 (bs, 1H, H-2′), 7.34 (d, *J* = 8.2 Hz, 1H, H-5), 7.47 (bs, 1H, H-3′), 7.83 (t, *J* = 7.8 Hz, 1H, H-8), 8.40 (d, *J* = 8.1 Hz, 1H, H-4), 8.44 (d, *J* = 8.4 Hz, 1H, H-7), 8.50 (d, *J* = 7.1 Hz, 1H, H-9). ^13^C NMR (150.9 MHz, DMSO-d_6_), δ (ppm): 10.2 (CH_3_-4′), 21.7 (CH_3_-4″), 30.2 (CH-4″), 33.9 (CH_2_-3″), 53.2 and 53.3 (CH_2_-2″ and CH_2_-6″), 104.0 (CH-2′), 114.6 (C-3a), 115.1 (CH-5), 122.7 (C-9a), 125.5 (C-6a), 125.9 (CH-8), 128.7 (C-9b), 131.1 (CH-9), 131.4 (CH-7), 132.8 (CH-4), 134.2 (C-4′), 141.9 (CH-3′), 157.2 (C-6), 159.9 (CO-3), 160.4 (CO-1), 171.0 (CO-5′). ^15^N NMR (60.8 MHz, DMSO-d_6_), δ (ppm): 73.6 (N-1″), 223.6 (N-2). HRMS-ESI (*m*/*z*): [M + Na]^+^ for C_23_H_22_N_2_NaO_5_, calcd. 429.1421, found 429.1425 [67].

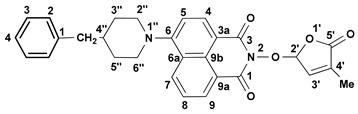

*2-(4-Methyl-5-oxo-2,5-dihydro-furan-2-yloxy)-6-(4-benzyl-piperidin-1-yl)-benzo[de]isoquinoline-1,3-dione (***3***):* **SL-21.** Orange crystals, m.p. 182–184 °C (CHCl_3_). Yield: 58% (1.67 g). Anal. Calcd. for C_29_H_26_N_2_O_5_ (482.54): C, 72.19; H, 5.43; N, 5.81%. Found: C, 72.06; H, 5.33; N, 5.93%. FTIR (KBr, ν_max_): 2911, 2794, 1784, 1718, 1684, 1584, 1452, 1364, 1235, 1177, 1026 cm^−1^. ^1^H NMR (600 MHz, DMSO-d_6_), δ (ppm): 1.58 (dd, *J* = 22.6, 11.2 Hz, 1H, CH_2_-3″A), 1.77 (d, *J* = 11.3 Hz, 2H, CH_2_-3″B and CH-4″), 1.91 (bs, 3H, CH_3_-4′), 2.65 (d, *J* = 6.5 Hz, 2H, CH_2_-4″), 2.87 (t, *J* = 12.1 Hz, 1H, CH_2_-2″A), 3.56 (d, *J* = 11.6 Hz, 1H, CH_2_-2″B), 6.60 (bs, 1H, H-2′), 7.21 (t, *J* = 7.2 Hz, 1H, H-4Ph), 7.24 (d, *J* = 7.3 Hz, 2H, H-2Ph), 7.31 (d, *J* = 7.2 Hz, 1H, H-5), 7.32 (t, *J* = 8.0 Hz, 2H, H-3Ph), 7.46 (bs, 1H, H-3′), 7.82 (t, *J* = 7.9 Hz, 1H, H-8), 8.39 (d, *J* = 8.2 Hz, 1H, H-4), 8.42 (d, *J* = 8.3 Hz, 1H, H-7), 8.49 (d, *J* = 7.2 Hz, 1H, H-9). ^13^C NMR (150.9 MHz, DMSO-d_6_), δ (ppm): 10.3 (CH_3_-4′), 31.8 (CH_2_-3″), 37.3 (CH-4″), 42.3 (CH_2_-4″), 53.2 and 53.3 (CH_2_-2″ and CH_2_-6″), 104.0 (CH-2′), 114.7 (C-3a), 115.1 (CH-5), 122.7 (C-9a), 125.5 (C-6a), 125.9 (CH-4Ph), 126.0 (CH-8), 128.2 (CH-3Ph), 128.7 (C-9b), 129.1 (CH-2Ph), 131.2 (CH-9), 131.4 (CH-7), 132.8 (CH-4), 134.2 (C-4′), 140.2 (C-1Ph), 141.9 (CH-3′), 157.1 (C-6), 160.0 (CO-3), 160.5 (CO-1), 171.1 (CO-5′). ^15^N NMR (60.8 MHz, DMSO-d_6_), δ (ppm): 74.2 (N-1″), 223.4 (N-2). HRMS-ESI (*m*/*z*): [M + Na]^+^ for C_29_H_26_N_2_NaO_5_, calcd. 505.1734, found 505.1738 [67].

### 3.5. Fungal Strain Cultivation and Experimental Design

The fungal strains were first inoculated and grown on potato dextrose agar (PDA) medium at 28 °C for 5 days. Stock solutions of SL mimics at a concentration of 10 mM were prepared by dissolving 1.5 mg of GR24, 2 mg of SL20, 2.5 mg of SL21, 3.43 mg of SL26, and 3.88 mg of SL27 in 500 µL of acetone. GR24 and SL mimics were tested by dissolving them in water agar 1.8%, with the final medium being poured into Petri dishes. The final concentrations of GR24 and SL mimics dissolved into water agar were 5 × 10^−6^ M (C1), 10^−5^ M (C2), and 5 × 10^−5^ M (C3). The acetone concentration in the final solution was 0.5% (*v*/*v*). Water agar of 1.8% and water agar embedded with 0.5% (*v*/*v*) acetone were used as controls, with GR24 as a chemical reference structure. The inoculated Petri dishes were kept for 3 days at 28 °C in an incubator MIR-154-PE (PHCBi Panasonic, Osaka, Japan). The developed colonies were macroscopically investigated by measuring the diameter. The images were collected on day 3 of shooting by examination in brightfield mode using a Leica DM 1000 LED microscope (Leica Microsystems, Wetzlar, Germany) equipped with an ICC50W digital camera. The number of hyphal branches of different orders was determined on each primary branch, from the second to the maximum, starting from the apical end of the youngest hyphae. The selected distance for hyphae counting was 2000 µm, where the maximum hyphae order could be observed for all variants. The observations focused on the number and the arrangement of the hyphae. Statistical analysis was performed using IBM^®^ SPSS^®^ Statistics version 26 (IBM, Armonk, NY, USA). For the statistical analysis, ten hyphae from three replicates for each variant were used. One-way ANOVA was used to determine whether there are significant differences between test groups.

## 4. Conclusions

Two new affordable bioactive SL mimics **7** and **10** containing a halo-substituted 1,8-naphthalimide ring connected through an ether link to a bioactive furan-2-one unit have been obtained and fully characterized. They have been further used in alternative synthetic routes towards the previously reported SL mimics **2** and **3.** All investigated compounds (**2**, **3**, **7**, and **10**) present fluorescence properties and interesting biological activity. Particularly, SL mimics **2** and **3**, bearing piperidine-substituents on the 1,8-naphthalimide ring, exhibit emission maxima in DCM at 522 nm and 520 nm, respectively; high Stokes shifts; and convenient quantum fluorescence yields. SL mimics **2**, **3**, 7, and **10** showed biological activity similar to GR24 on selected phytopathogen strains. The specific behavior depends on the strain, SL mimic characteristics, and concentration, as well as on hyphal order. These characteristics make them, SL mimic **2** (SL20) in particular, convenient for sustainable agriculture and bio-imaging applications. The specific behavior depends on the strain, SL mimic characteristics, and concentration, as well as hyphal order. The fluorescence after the hydrolysis of the bioactive D-ring is preserved. Therefore, it could be further used to investigate the temporal distribution of the fluorescent bioactiphore probe in the plant or fungal cell.

## Data Availability

The data are included in the main manuscript and Appendix A.

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
