# Peer review of "Synthesis and Biological Properties of Fluorescent Strigolactone Mimics Derived from 1,8-Naphthalimide"

_molecules, 2024, doi:10.3390/molecules29102283_

Round 1

Reviewer 1 Report

Comments and Suggestions for Authors

1.  The authors should improve the introduction part, including list the important structure of Gr24 analogues and recent development..

2. The author should prepare more analogues based this structure and investigate the SAR relationship.

Comments on the Quality of English Language

no

Author Response

Comments 1. The authors should improve the introduction part, including list the important structure of Gr24 analogues and recent development.

Response 1. Thank you for your comment. In the Introduction, we include a list that describes the strigolactone analogues and mimics, starting with GR24, and recent development - L71-L87.

Comments 2. The author should prepare more analogues based this structure and investigate the SAR relationship.

Response 2. Thank you for your comment. We are continuing our work on SL mimics and analogues based on this structure. The investigation of the SAR relationship is being worked on, and this will be part of a forthcoming paper we are preparing. 

Reviewer 2 Report

Comments and Suggestions for Authors

The manuscript “Synthesis, Spectroscopic and Biological Properties of Fluorescent Strigolactone Mimics derived from 1,8-Naphthalimide” presented to Molecules is detailed, but in my humble opinion it presents too many aspects already published by the authors previously. However, I emphasize that the main novelty is the biological activity of compounds 2 and 3, although they are synthetically described, their bioactive properties were unknown. For this reason, I recommend its Reconsider after major, which should include the following:

1) An illustrative figure with the structure of natural SLs and the ring systems A, B, C and D that they describe should be included in the introduction. Include the biological properties described for this family of compounds.

2) Section 2 Results, should be changed to 2. Results and Discussion

3) Reverse the synthesis steps 2 and 3, do not make it a new synthetic protocol. Delete that expression. How does reversing the synthesis stages of 2 and 3 affect the overall performance of each process with respect to what is described in ref. 70? Is it significant (increase in overall performance by more than 25-30%, for 3-stage routes)?

4) Include the yields of each reaction in schemes 1 and 2. Include these yields in the text of the results section.

5) Considerably improve the quality of the image in fig. 2, especially the structure of compound 7.

6) In relation to the spectroscopic properties, in my opinion, in this work they do not represent an advance, since the target compounds of the work were already studied and their spectroscopic properties were described in references 69 and 70. To include them again is to repeat information. The comps. 7 and 10, would be the “new” but their properties are significantly inferior to those already described. I recommend removing this section from the main text and including it in the SI, making a brief mention that the spectroscopic properties of these compounds are included in the SI and compared with already known compounds.

7) The change of nomenclature in section 2.3 (biological activity) seems to me to be a distraction for readers. The change must be made to the numbering that the manuscript presents from the beginning.

8) In section 2.3 compound GR24 is used as a control. The bibliographic background to be used as a control and its structure must be specified.

9) In section 3.1, the authors describe that they made unambiguous assignments for 1H, 13C and 15N NMR using COZY, HSQC and HMBC spectra, but all of these spectra are not in the SI, please include all 2D -NMR

10) Conclusions: They must be deeply reformulated, avoiding the use of the word “Novel”, since the best results are from compounds 2 and 3 that are already known, the only novelty in this work is the biological evaluation of these two compounds.

11) Remove "spectroscopic" from the title

Comments on the Quality of English Language

They must check the grammar in all the work. Adding to these:

1. Line 96, the word “plannat” must be corrected (it does not exist in English, it does not appear in the dictionary).

2. Line 55: synthesizing of simplified molecules should be changed to synthesizing simplified molecules.

3. Line 68: the word "evoluate" needs to be modified (it does not appear in the dictionary), perhaps evolved?

Author Response

The manuscript “Synthesis, Spectroscopic and Biological Properties of Fluorescent Strigolactone Mimics derived from 1,8-Naphthalimide” presented to Molecules is detailed, but in my humble opinion it presents too many aspects already published by the authors previously. However, I emphasize that the main novelty is the biological activity of compounds 2 and 3, although they are synthetically described, their bioactive properties were unknown.

Thank you very much for taking the time to review this manuscript. Please find the detailed responses below and the corresponding revisions/corrections highlighted/in track changes in the re-submitted filer

For this reason, I recommend its Reconsider after major, which should include the following:

Comments 1. An illustrative figure with the structure of natural SLs and the ring systems A, B, C and D that they describe should be included in the introduction. Include the biological properties described for this family of compounds.

Response 1. We included Figure 1 in the Introduction section, describing the naturally occurring canonical strigolactone structures, with A, B, C and D rings, from the family (+)-strigol and (−)-orobanchol. We described the biological properties of SLs as exo- and endo-signals – L37-L49. We added a paragraph mentioning the role of C (open or closed) and D rings, connected via an enol ether bridge for SLs biological properties.

Comments 2.  Section 2 Results, should be changed to 2. Results and Discussion

Response 2.  We have changed the title of Section 2 in Results and Discussion in the paper.

Comments 3. Reverse the synthesis steps 2 and 3, do not make it a new synthetic protocol. Delete that expression. How does reversing the synthesis stages of 2 and 3 affect the overall performance of each process with respect to what is described in ref. 70? Is it significant (increase in overall performance by more than 25-30%, for 3-stage routes)?

Response 3. Thank you for your comment. We agree with you. It is not a new synthetic protocol, and we deleted this expression. We only prepared two others new 1,8-naphthalimide-derived SL mimics 7 and 10 and we used them to obtain the already reported compounds 2 and 3 via an alternative procedure. The final yields in compounds 2 and 3 synthesized according to the already described procedure [70] and the alternative procedure presented in this paper are quite similar. This alternative procedure proves the versatility of the initial synthetic procedure, enabling future syntheses based on more complicated synthons.

Comments 4. Include the yields of each reaction in schemes 1 and 2. Include these yields in the text of the results section.

Response 4. We included the yields on each step in Schemes 1 and 2 and in the experimental section.

Comments 5. Considerably improve the quality of the image in fig. 2, especially the structure of compound 7.

Response 5.  We have improved the image quality in fig. 2 (renumbered as figure 3).

Comments 5. In relation to the spectroscopic properties, in my opinion, in this work they do not represent an advance, since the target compounds of the work were already studied and their spectroscopic properties were described in references 69 and 70. To include them again is to repeat information. The comps. 7 and 10, would be the “new” but their properties are significantly inferior to those already described. I recommend removing this section from the main text and including it in the SI, making a brief mention that the spectroscopic properties of these compounds are included in the SI and compared with already known compounds.

Response 6. Thank you for your comment. We agree with you. The spectroscopic properties of the SL mimics 7 and 10 are inferior to those already reported for SL mimics 2 and 3, as mentioned in the paper. Our idea was to present a comparative study on the spectroscopic properties of synthesized mimics derived from 1,8-naphthalic anhydrides. As you recommended, we removed this section from the main text and included a paragraph about the unreported photophysical properties of compounds 7 and 10. We placed the comparative data on spectroscopic properties of SL mimics 2 and 3 in the Supplementary Material.

Comments 7.  The change of nomenclature in section 2.3 (biological activity) seems to me to be a distraction for readers. The change must be made to the numbering that the manuscript presents from the beginning.

Response 7. Thank you for your comment. This change was related to further using the strigolactone mimics as fluorescent probes or replacers of the strigolactone in laboratory tests related to various biological properties of these strigolactone mimics. This change is intended to support readers who are not focused on the strigolactone analogues or mimics synthesis but on the biological characteristics of different structures. Such synthetic compounds are designed to be used in various biological systems and need a given name, a laboratory code (e.g., GR24), or a common name (contalactone, i.e., contaminant lactone of GR24). We introduced Chart 1. Investigated strigolactone mimics, to avoid reader confusion. Here, we mention that other groups also made such changes in their papers reporting synthetic strigolactone derivatives — from numbering to given names. We randomly cited three papers: Bhattacharya, C., Bonfante, P., Deagostino, A., Kapulnik, Y., Larini, P., Occhiato, E. G., Prandi, C. & Venturello, P. (2009). A new class of conjugated strigolactone analogues with fluorescent properties: synthesis and biological activity. Organic & biomolecular chemistry7(17), 3413-3420; Rasmussen, A., Heugebaert, T., Matthys, C., Van Deun, R., Boyer, F. D., Goormachtig, S., Stevven, C. & Geelen, D. (2013). A fluorescent alternative to the synthetic strigolactone GR24. Molecular plant6(1), 100-112; Fornier, S. D., de Saint Germain, A., Retailleau, P., Pillot, J. P., Taulera, Q., Andna, L., ... & Boyer, F. D. (2022). Noncanonical strigolactone analogues highlight selectivity for stimulating germination in two Phelipanche ramosa populations. Journal of Natural Products85(8), 1976-1992.

Comments 8. In section 2.3 compound GR24 is used as a control. The bibliographic background to be used as a control and its structure must be specified.

Response 8. Thank you for your comment. The bibliographic background for using this SL analogue with proven effect on branching of plant pathogenic fungi was included. We also mentioned that this SL analogue was developed from strigol as a basic blueprint and include a tricyclic lactone system (A, B and C rings) connected through an enol-ether bond to a furan-2-one ring (D)

Comments 9. In section 3.1, the authors describe that they made unambiguous assignments for 1H, 13C and 15N NMR using COZY, HSQC and HMBC spectra, but all of these spectra are not in the SI, please include all 2D -NMR

Response 9. We have included these spectra in the SI part.

Comments 10. Conclusions: They must be deeply reformulated, avoiding the use of the word “Novel”, since the best results are from compounds 2 and 3 that are already known, the only novelty in this work is the biological evaluation of these two compounds.

Response 10. We reformulate the conclusion. We avoid to use novel related to the compounds.

Comments 11.  Remove "spectroscopic" from the title

Response 11. We remove the word "spectroscopic" from the title.

Comments on the Quality of English Language

They must check the grammar in all the work. Adding to these:

  1. Line 96, the word “plannat” must be corrected (it does not exist in English, it does not appear in the dictionary).

Response 1a. We have corrected: “plant” instead of “plannat”.  

  1. Line 55: synthesizing of simplified molecules should be changed to synthesizing simplified molecules.

Response 2a . We’ve corrected. Thank you!

  1. Line 68: the word "evoluate" needs to be modified (it does not appear in the dictionary), perhaps evolved?

Response 3a. We corrected. Thank you.

Reviewer 3 Report

Comments and Suggestions for Authors

The authors report the synthesis, spectroscopic properties, and biological activity of fluorescent stringolactone (SL) mimics. In my opinion, this article deserves publication in Molecules after minor revisions.

Main point:

In the manuscript, the compounds that are derivatives of 1,8-naphthalimide are described as new SL mimics (lines 22, 24, 96 , …., 290, 536). However, two of the four compounds have already been described by the same authors in ref. 70 (see lines 80-84). They were prepared by ‘a new synthetic protocol’ (line 98) consisting in the inversion of two reaction steps. I think it should be clearer what are the original contributions of this article and what are not. This should include a discussion of the improvements made. In general, schemes reporting a synthesis should show the yields. It would be useful to add a scheme showing the previous synthetic approach with yields for comparison.

Identifying each structure with a number and an alphanumeric label, and using them simultaneously or alternately, creates confusion: it is preferable to use only one system.

Other points to be considered:

Wrong fonts:

[de], H, N, in the chemical names should be in italics (lines 79, 89, 125, 132, 159, ……)

Numbers indicating a structure should be in bold (lines 79, 82, 83, 89, 90, 99, 268, 269……)

Lines 279 and 280: figure 4 is repeated.

Lines 546-548: this sentence should be rephrased.

References: Journal titles are not abbreviated.

Ref. 61 and 69: some parts are in capital letters.

Author Response

The authors report the synthesis, spectroscopic properties, and biological activity of fluorescent strigolactone (SL) mimics. In my opinion, this article deserves publication in Molecules after minor revisions.

Thank you very much for taking the time to review this manuscript and for the appreciation. Please find the detailed responses below and the corresponding revisions/corrections highlighted/in track changes in the re-submitted file.

 Main point:

Comments 1. In the manuscript, the compounds that are derivatives of 1,8-naphthalimide are described as new SL mimics (lines 22, 24, 96 , …., 290, 536). However, two of the four compounds have already been described by the same authors in ref. 70 (see lines 80-84). They were prepared by ‘a new synthetic protocol’ (line 98) consisting in the inversion of two reaction steps. I think it should be clearer what are the original contributions of this article and what are not. This should include a discussion of the improvements made. In general, schemes reporting a synthesis should show the yields. It would be useful to add a scheme showing the previous synthetic approach with yields for comparison.

Response 1. We rephrased the last part of Introduction and the Conclusions in order to clearly present the original contribution of this paper which are: two new prepared and completely characterized compounds 7 and 10, the possibility to use them in the preparation of already reported SL mimics 2 and 3 by alternative routes and the biological study about SL mimic compounds 2, 3, 7 and 10. We included the yields in the Schemes 1 and 2.

Comments 2. Identifying each structure with a number and an alphanumeric label, and using them simultaneously or alternately, creates confusion: it is preferable to use only one system.

Response 2. Thank you for your comment. This double identification, with a number and an alphanumeric label, is related to further using the strigolactone mimics as fluorescent probes or replacers of the strigolactone in laboratory tests related to various biological properties of these strigolactone mimics. This double idnetification is intended to support readers who are not focused on the strigolactone analogues or mimics synthesis but on the biological characteristics of different structures. Such synthetic compounds are designed to be used in various biological systems and need an alphanumeric label. We introduced Chart 1. Investigated strigolactone mimics, to avoid reader confusion.  We mention that other groups also use double identification in their papers reporting synthetic strigolactone derivatives —numers and alphanumeric characters. We randomly cited three papers: Bhattacharya, C., Bonfante, P., Deagostino, A., Kapulnik, Y., Larini, P., Occhiato, E. G., Prandi, C. & Venturello, P. (2009). A new class of conjugated strigolactone analogues with fluorescent properties: synthesis and biological activity. Organic & biomolecular chemistry7(17), 3413-3420; Rasmussen, A., Heugebaert, T., Matthys, C., Van Deun, R., Boyer, F. D., Goormachtig, S., Stevven, C. & Geelen, D. (2013). A fluorescent alternative to the synthetic strigolactone GR24. Molecular plant6(1), 100-112; Fornier, S. D., de Saint Germain, A., Retailleau, P., Pillot, J. P., Taulera, Q., Andna, L., ... & Boyer, F. D. (2022). Noncanonical strigolactone analogues highlight selectivity for stimulating germination in two Phelipanche ramosa populations. Journal of Natural Products85(8), 1976-1992.

Other points to be considered:

Wrong fonts:

Comments 3. [de], H, N, in the chemical names should be in italics (lines 79, 89, 125, 132, 159, ……)

Numbers indicating a structure should be in bold (lines 79, 82, 83, 89, 90, 99, 268, 269……)

Response 3. we’ve corrected the above observation.

Comments 4. Lines 279 and 280: figure 4 is repeated.

Response 4. Thank you, we corrected.

Comments 5. Lines 546-548: this sentence should be rephrased.

Response 5. Thank you. We modified.

Comments 6. References: Journal titles are not abbreviated.

Ref. 61 and 69: some parts are in capital letters.

Response 6. We’ve corrected all references and abbreviated journal titles. Thank you!

Round 2

Reviewer 1 Report

Comments and Suggestions for Authors

The authors have followed the suggestion and made the revision.

Reviewer 2 Report

Comments and Suggestions for Authors

The revised manuscript “Synthesis and Biological Properties of Fluorescent Strigolac-2 tone Mimics derived from 1,8-Naphthalimide” has been corrected in detail incorporating all the comments made in the first revision. Therefore, I support its publication in Molecules.